# Nutrition, Flavor, and Microbial Communities of Two Traditional Bacterial Douchi from Gansu, China

**DOI:** 10.3390/foods13213519

**Published:** 2024-11-04

**Authors:** Haijun Qiao, Yaping Li, Fengyun Cui, Weibing Zhang, Zhongming Zhang, Huifeng Li

**Affiliations:** 1College of Science, Gansu Agricultural University, Lanzhou 730070, China; qiaohj1999@126.com; 2College of Food Science and Engineering, Gansu Agricultural University, Lanzhou 730070, China; kitty_xl@126.com; 3Science and Technology Research Center of China Customs, Beijing 100026, China; cuifyciq@163.com; 4Beijing Key Laboratory of Energy Conversion and Storage Materials, College of Chemistry, Beijing Normal University, Beijing 100875, China; lihuifeng@bnu.edu.cn

**Keywords:** traditional bacterial douchi, nutrition, flavor, correlation

## Abstract

Douchi has attracted attention for its unique taste and rich health functions. This study investigated the nutrition, flavor and correlation between the flavor and microorganisms of two traditional bacterial douchi from the province of Gansu in northwest China. The findings reveal significant variations in nutrition, flavor compounds, and the microbiota between Longnan and Qingyang douchi. Three dominant bacterial genera (*Carnobacterium*, *Ignatzschineria*, and *Bacillus*) and one dominant bacterial genus (*Pichia*) were found in the QY douchi, while four bacterial genera (*Bacillus*, *Ignatzschineria*, *Proteus*, and *Providencia*) and three fungal genera (*Pichia*, *Candida*, and *Rhodosporidium*) were dominant in samples of the LN douchi. For flavor substances, a total of 48 volatile components were detected in Longnan douchi and 41 in Qingyang douchi. Using the relative odor activity value (ROAV), we identified five key flavor compounds in Longnan douchi and four key flavor compounds in Qingyang douchi. The correlation analysis showed that there were certain positive or negative correlations between the key microorganisms and the flavor of the two traditional bacterial douchi. The results of this study can serve as a theoretical reference for improving the quality and flavor of traditional douchi.

## 1. Introduction

There are many types of fermented bean products, including natto, miso, tofu, douchi, sufu, cheonggukjang, doenjang, kanjang, meju, and tempeh [1]. Many microorganisms are involved in the production process of fermented bean products, and some bean products require additional inoculation of specific microorganisms for fermentation. During the fermentation process, microorganisms can produce various enzymes, including amylase, protease, and lipase, which play an important role in the quality and flavor formation of fermented bean products [2,3]. The types of microorganisms in traditional fermented bean products are influenced by various factors, including fermentation materials, processing personnel, production environment, etc. [4].

Douchi is a traditional fermented bean product that is widely used as a condiment in China due to its unique flavor and taste [1]. Douchi is generally divided into two types: fungal-type douchi and bacterial-type douchi. According to the difference in inoculated microorganisms, fungal-type douchi can be divided into *Aspergillus*-type douchi, *Mucor*-type douchi, and *Rhizopus*-type douchi. During the fermentation of bacterial-type douchi with no artificial inoculation of microorganisms, naturally occurring microorganisms, which are mainly bacteria with a small amount of fungi, play a role in fermentation. Douchi is fermented by microorganisms that help to retain the inherent nutritional components of soybeans [2]. Douchi is rich in nutrients and beneficial amino acids, as well as various vitamins, minerals, and physiologically active substances required by the human body. It also has medicinal value and a unique flavor [3]. The formation of flavor compounds in douchi is mainly through three main pathways that are closely related to the microbial community. First, the microbial community directly metabolizes and processes large compounds into smaller flavor components. Second, the microbial community produces enzymes related to metabolism that degrade large compounds into small flavor components. In addition, microorganisms themselves can produce precursors of flavor compounds [1,2]. Therefore, the flavor of douchi mainly comes from aroma components (volatile compounds) and flavor substances (mainly free amino acids). The direct contribution of FAAs to flavor is limited, but they are precursors for the formation of other flavor compounds [4]. FAAs can decarboxylate to form amines and carbon monoxide and deaminate to form ammonia and α- keto acid (further converted into flavor compounds such as aldehydes, alcohols, and acids) [5]. For example, tryptophan generates indole pyruvate through the action of transaminases, which further decompose and metabolize to benzaldehyde [6].

Longnan and Qingyang douchi are two typical bacterial-type douchi from Gansu Province in northwest China, which are made of soybeans. These two kinds of douchi are very popular with the local people. There have been numerous reports on the nutrition and flavor of *Aspergillus*-type douchi, *Mucor*-type douchi, and *Rhizopus*-type douchi [7,8]. However, there are few reports on the nutrition and flavor of bacterial-type douchi [9,10]. Therefore, this study focused on the traditional fermented bacterial-type douchi in Longnan and Qingyang and compared their nutritional components and flavor substances. In addition, it analyzed the correlation between flavor and microorganisms to provide theoretical support for the quality control and evaluation of bacterial-type douchi.

## 2. Materials and Methods

### 2.1. Sample Collection and Preprocessing

Douchi samples were collected from the finished products of homemade douchi made by villagers in the cities of Longnan and Qingyang in Gansu. A total of 60 douchi samples (30 from six villagers in Longnan cities, 30 from six villagers in Qingyang cities) were put in an ice box and transported to our lab. Subsequently, five samples from one villager were mixed into a composite sample and placed in sterile tubes and then stored at –18 °C.

### 2.2. Determination of Nutrients in the Two Types of Douchi

#### 2.2.1. Determination of Basic Physicochemical Indices in the Two Types of Douchi

The determination methods for moisture, fat, protein, salt, total acid, and amino acid nitrogen followed Chinese Standards GB 5009.3-2016, GB 5009.6-2016, GB 5009.5-2016, GB 5009.44-2016, GB 12456-2021, and GB5009.235-2016, respectively. These standards can be found online at foodmate (http://down.foodmate.net/standard/, accessed on 20 August 2024).

#### 2.2.2. Determination of Free Fatty Acids (FFAs) in the Two Types of Douchi

The determination methods for FFAs followed Chinese Standard GB 5009.168-2016.

#### 2.2.3. Determination of Free Amino Acids (FAAs) in the Two Types of Douchi

The determination methods for FAAs followed Chinese Standard GB 5009.124-2016.

The essential amino acid pattern of FAO/WHO was used as the standard pattern, and the ratio coefficient of the amino acid (RC) and the score of ratio coefficient of the amino acid (SRC) of the two types of douchi were calculated with reference to the method of Zhu et al. [11] to evaluate the amino acid patterns in the two types of douchi. The ratio of the amino acid (RAA) refers to the ratio of amino acid content to the standard amino acid content in a certain amount of food, and the RC is used to determine the limiting amino acid and calculate the fortification amount of the limiting amino acid. The SRC is an important index for judging the nutritional value of food. If RC = 1, the amino acid composition in the sample is consistent with the standard amino acid pattern; if RC > 1, the amino acid is relatively surplus; if RC < 1, the amino acid is relatively insufficient, and the amino acid with the minimum RC value is the first limiting amino acid of the sample [12]. The smaller the SRC, the lower the nutritional value. The closer the SRC is to 100, the higher the nutritional value is. If the SRC is equal to 100, it indicates that the proportion of the essential amino acid composition is consistent with the pattern spectrum in the sample.

The formulas are as follows:(1)$$RAA=\frac{The\;content\;of\;a\;certain\;amino\;acid\;in\;douchi}{The\;amino\;acid\;content\;in\;the\;standard\;mode}$$
(2)$$RC=\frac{RAA}{Mean\;value\;of\;RAA}$$
(3)$$SRC=100-\frac{The\;standard\;deviation\;of\;RAA}{Mean\;value\;of\;RAA}\times 100$$

### 2.3. Determination and Analysis of Volatile Flavor Compounds in the Two Types of Douchi

#### 2.3.1. Determination of Volatile Flavor Compounds

The determination method is based on a reported method with slight modification [13]. The homogenized douchi sample was weighed (3.0 g) in a 15 mL sample bottle with a PTFE spacer, Na_2_SO_4_ was added, and distilled water was added to dissolve the flavor compounds fully. Then, the bottle was covered and sealed, stirred at 60 °C with magnetic force, and inserted into an activated DVB/CAR/PDMS-coated solid-phase microextraction (SPME) extractor head (activated at 270 °C for 30 min). The fiber head was exposed to the sample bottle for constant temperature adsorption in the headspace for 30 min and then inserted into the sample inlet at 230 °C for analysis for 5 min. The same sample was measured in parallel three times.

Mass spectrometry conditions: transmission line temperature, 200 °C; interface temperature, 250 °C; detector voltage, 350 V; emission current, 150 µA; scanning mode, full scan. Chromatographic conditions: DB-5 capillary column, 60 m × 0.25 mm × 0.25 μm. Heating program: initial temperature of 40 °C for 2 min, the temperature was then raised at a rate of 3 °C/min to 180 °C, maintained for 5 min with a helium flow rate of 1.0 mL/min.

Qualitative analysis was conducted using the Wiley spectral library. The volatiles were screened out based on the principles of positive and negative matching degree > 80% and high reproducibility of chromatographic peaks. The relative content of the tested components in fermented douchi was calculated using the peak-area normalization method.

#### 2.3.2. Identification of the Key Flavor Compounds in the Two Types of Douchi by ROAV

In referring to the method of Cai et al. [14], the ROAV was used to determine the key flavor compounds in two types of douchi samples. In the two types of douchi, the ROAVs of the other compounds (a) were determined using the following formula:(4)$${ROAV}_{a}=\frac{C\%a}{C\%max}\times \frac{Tmax}{Ta}\times 100$$
where C%a and C%max denote the relative content of each flavor compound and the flavor compound with the largest contribution to the aroma, respectively, and Ta and Tmax denote the odor threshold of each flavor compound and the flavor compound with the largest contribution to the aroma, respectively.

From the above formula, it can be seen that for the main flavor compounds of the douchi sample, ROAV ≥ 1. The compounds with 0 < ROAV < 1 have a modifying effect on the flavor of the sample.

### 2.4. Characterization of Microbiota and Their Correlation with Flavor Compounds in the Two Types of Douchi

Here, 16s and ITS high-throughput sequencing were used to study the major microorganisms in the two types of douchi. Total genomic DNA extraction and amplification were performed as in a previous study [8]. Sequencing was performed using the standard protocols of Beijing Biomarker Technologies Co., Ltd. (Beijing, China). After sequencing, microbiome bioinformatic analysis was performed with QIIME2 (https://qiime2.org/) according to the official tutorials. The correlation between bacterial genera and characteristic VFCs was also analyzed on the OmicShare online platform “https://www.omicshare.com/ (accessed on 20 August 2024)”.

### 2.5. Data Analysis

The full-text data were processed and analyzed using SPSS (version 19.0, IBM, Inc., Armonk, NY, USA), and the data were represented in the form of mean ± standard deviation. Pearson correlation analysis was performed using Origin (version 9.0, OriginLab, Northampton, MA, USA). A significant correlation was defined as a correlation >0.6 and *p* < 0.05.

## 3. Results and Analysis

### 3.1. Analysis of Nutritional Components of Longnan and Qingyang Douchi

#### 3.1.1. Analysis of Basic Physicochemical Indicators of the Two Types of Douchi

The basic nutritional components of Longnan and Qingyang douchi are shown in Table 1. There was no significant difference in total acids between the two types of douchi (*p* < 0.05), but there were significant differences in moisture, fat, amino acid nitrogen, salt, and protein content (*p* < 0.05). The protein, amino acid nitrogen, and salt content in Longnan douchi were higher, namely, 16.5%, 32.7%, and 29.2% higher than those in Qingyang douchi, respectively; Qingyang douchi had a higher content of moisture and fat, which were 8.8%and 77.2% higher than in Longnan douchi, respectively.

During the fermentation process of douchi, microorganisms in it degrade macromolecular substances, for example, by reducing sugars into small-molecule monosaccharides, disaccharides, etc., through decomposition, metabolism, and other transformation actions, increasing the nutrition of the raw material while improving its bioavailability. These small-molecule substances also participate in the formation of the nutritional structure and special flavor substances of douchi [7], thereby determining its quality. Overall, the nutritional composition of douchi was mainly determined by the microorganisms contained and the enzymes they produce. The differences in the basic nutrient composition of the two different douchi were mainly due to the differences in the types and content of the microorganisms and enzymes within.

#### 3.1.2. Analysis of FFAs in the Two Types of Douchi

The results of FFA content determination in Longnan and Qingyang douchi are shown in Table 2. A total of 16 FFAs were detected in Longnan douchi, whereas only 14 were detected in Qingyang douchi. Among them, there were the most types of saturated fatty acids (SFAs) and the highest content of polyunsaturated fatty acids (PUFAs). Longnan and Qingyang douchi contained seven types of SFAs, namely, myristic acid, palmitic acid, heptadecanoic acid, stearic acid, arachidic acid, behenic acid, and lignoceric acid, accounting for 14.7% and 14.0% of the total FFA content, respectively. Both types of douchi had the highest content of palmitic acid and the next highest content of stearic acid among the SFAs. The two types of douchi contained five monounsaturated fatty acids (MUFAs), namely, *cis*-10-heptadecenoic acid, oleic acid, *cis*-11-eicosenoic acid, erucic acid, and *cis*-15-tetracosenic acid, accounting for 19.2% and 20.1% of the total FFAs content, respectively. Both types of douchi had oleic acid as the highest content among MUFAs. The two types of douchi contained four PUFAs: linoleic acid, linolenic acid, docosadienoic acid, and *cis*-11, 14-eicosadienoic acid, accounting for 66.1% and 65.9% of the total FFA content, respectively. The PUFAs in both types of douchi with the highest content were linoleic acid followed by linolenic acid.

The content of palmitic acid, *cis*-10-heptadecenoic acid, and erucic acid in Longnan douchi was significantly higher than in Qingyang douche (*p* < 0.05), which were 10.6%, 33.3%, and 250.0% higher, respectively. *Cis*-15-tetracosenic acid and docosadienoic acid were not detected in Qingyang douchi. The content of arachidic acid and lignoceric acid in Qingyang douchi was significantly higher than in Longnan douchi (*p* < 0.05), which were 38.1% and 83.3% higher, respectively. There was no significant difference in the content of stearic acid, behenic acid, *cis*-11-eicosenoic acid, or *cis*-11, 14-eicosadienoic acid between the two types of douchi (*p* > 0.05).

The main components of common fermented soybean condiments including proteins, fats, sugars, etc., are degraded by a series of enzymes such as proteases, amylases, cellulases, and lipases provided by microorganisms, producing a variety of metabolites such as fatty acids, amino acids, and organic acids, which contribute to the unique nutrition, flavor, taste, and functionality of douchi [15]. The intake of FFAs in food is closely related to human health. Some studies have found that the intake of foods with appropriate ratios of SFAs and PUFAs can reduce the morbidity and mortality of some diseases [16]. It was reported that a PUFA/SFA ratio that is less than 0.45 is unhealthy [17]. The PUFA/SFA values of Longnan and Qingyang douchi were 4.49 and 4.71, respectively, indicating that the ratios of FFAs in the Longnan and Qingyang douchi were reasonable and healthier.

#### 3.1.3. Analysis of FAAs in the Two Types of Douchi

The results of FAA composition and content of Longnan and Qingyang douchi are shown in Table 3. A total of 6 essential amino acids and 10 non-essential amino acids were detected in the two types of douchi, and the total concentration of FAAs in Longnan and Qingyang douchi was 19.31 g/100 g and 12.58 g/100 g, respectively. There were significant differences (*p* < 0.05) in the concentration of the 16 FAAs between the two types of douchi. The highest levels were glutamate detected in both Longnan and Qingyang douchi, with 6.13 g/100 g and 4.81 g/100 g, respectively. The second highest concentration was proline at 2.11 g/100 g and 1.68 g/100 g, respectively. The high content of glutamate in douchi reacts with salt to produce sodium glutamate, which plays an important role in the taste of douchi, this may also affect the final taste of douchi [8]. Among the 16 FAAs in the two types of douchi, except for tyrosine, the content of the other 15 FAAs in Longnan douchi was higher than that in Qingyang douchi. The concentration of umami amino acids (aspartate and glutamate) in Longnan and Qingyang douchi was the highest, with 6.97 g/100 g and 5.45 g/100 g, respectively, accounting for 36.1% and 43.3% of the total FAAs content of the two types of douchi. The concentration of sweet amino acids in the two types of douchi was 6.28 g/100 g and 3.68 g/100 g, respectively, accounting for 32.5% and 29.3% of the total amino acid content in the two types of douchi. The concentration of bitter amino acids in the two types of douchi was 4.68 g/100 g and 2.67 g/100 g, respectively, accounting for 24.2% and 21.2% of the total amino acid content in the two types of douchi.

FAAs are one of the main flavor substances in douchi, and their composition determines the sensory characteristics of douchi. The content of FAAs has a significant impact on the flavor and quality of fermented food. Among them, glutamate and aspartate are umami amino acids [18], whereas isoleucine, leucine, and arginine are bitter amino acids [19]. The combination of multiple FAAs has achieved the unique flavor and sensory characteristics of douchi.

As shown in Appendix A, the content of essential amino acids in Longnan douchi was higher than that in Qingyang douchi. However, the content of all types of essential amino acids in the two types of douchi was lower than the WHO/FAO-recommended amino acid standard pattern. Therefore, the amino acid ratio coefficient method was used to evaluate the protein nutritional value of the essential amino acids in the two types of douchi. As shown in Appendix A, the amino acid with the minimum RC in the two types of douchi was threonine, which was the first limiting amino acid for both types of douchi. The SRC of Qingyang and Longnan douchi were 78.51 and 79.51, respectively, indicating that both types of douchi had high nutritional value, and maybe Longnan douchi had higher protein nutritional value than Qingyang douchi.

### 3.2. Analysis of Volatile Flavor Compounds in the Two Types of Douchi

#### 3.2.1. Determination of Volatile Flavor Compounds

With the using of headspace solid-phase microextraction-gas chromatography-mass spectrometry (HS-SPME-GC /MS), the volatile compounds in the two kinds of traditional fermented douchi from Longnan and Qingyang were determined, and the relative content of flavor compounds was analyzed for significance. The results are shown in Table 4. A total of 77 volatile flavor compounds, including esters, alcohols, ketones, aldehydes, and terpenoids, were detected in the two types of douchi.

A total of 48 volatile compounds were detected in Longnan douchi, and the category with the largest number (15) was esters. The categories with high relative content were esters, pyrazines, and alcohols, and their relative contents were 33.32%, 25.55%, and 14.29%, respectively. The relative content of ligustrazine was the highest in Longnan douchi, and the relative content of 2,4,5-trimethylthiazole, 2,3-dimethylpyrazine, 2,3,5-trimethylpyrazine, ethyl octanoate, 1-octen-3-ol, ligustrazine, and methyl benzoate was significantly higher than that in Qingyang douchi (*p* < 0.05). Esters were the main category of compounds in Longnan douchi, mainly including ethyl isovalerate, ethyl butyrate, ethyl isobutyrate, ethyl acetate, etc. They mainly use organic acids and alcohols in douchi as precursors and generate esterification reactions with free fatty acids [20]. In addition, esters can also be extracted from raw materials, such as Baijiu.

A total of 41 volatile compounds were detected in Qingyang douchi, the most diverse category was ketones with 16 types. The categories with higher content were pyrazines, ketones, and alcohols, and their relative contents were 40.64%, 31.27%, and 8.90%, respectively. The relative content of 2,5-dimethylpyrazine in Qingyang douchi was the highest, and the relative content of isoamyl alcohol, 3-hydroxy-2-butanone, 2,5-dimethylpyrazine, 2,6-dimethylpyrazine, and acetophenone was significantly higher than that in Longnan douche (*p* < 0.05). During the fermentation process of Qingyang douchi, the main volatile compounds were ketones, which exhibit a fresh, grassy, and fatty flavor [21]. Fat oxidation and the Maillard reaction may contribute to the formation of ketones [22].

The flavor compounds that were shared and counted for a relatively large proportion of Longnan and Qingyang douchi were pyrazines and alcohols. Pyrazines were formed through a Maillard browning reaction or microbial metabolism [23]. Pyrazines have a low threshold and contribute significantly to unique flavors, such as nutty, barbecue, smoky, saucy, milky, and other good flavors [24,25]. Alcohols provide beneficial flavors to douchi such as sweet, nutty, and bread flavors, which can be formed through the Maillard reaction and fat oxidation [7].

#### 3.2.2. Identification of Key Flavor Compounds Using ROAV

In identifying the compound that contributes the most to the flavor of the douchi, when the concentration of the compound is fixed, the sensory threshold is inversely proportional to the degree of perceptual sensitivity; namely, the lower the compound threshold, the easier it is to perceive. When the sensory threshold of a compound is fixed, the concentration is directly proportional to the degree of perceptual sensitivity; namely, the higher the compound concentration, the easier it is to perceive [26].

The results of an analysis of the odor threshold of volatile compounds and ROAV calculations are shown in Table 5. In Longnan douchi, the relative content of ethyl isobutyrate was 4.36%, and the flavor threshold value was 0.1 μg/kg, which contributed the most to the overall flavor of Longnan douchi. The results show that there were five main flavor compounds (ROAV ≥ 1) in Longnan douchi, namely, ethyl isobutyrate, ethyl butyrate, isoamyl acetate, ethyl hexanoate, and 1-octen-3-ol, and there were three compounds that played a modifying role in the overall flavor of Longnan douchi (0.1 ≤ ROAV < 1), namely, ethyl heptanoate, 3-octanone, and 2,4,5-trimethylthiazole. In Qingyang douchi, 1-octen-3-ol, with a relative content of 1.38% and a flavor threshold of 1 μg/kg, contributed the most to the overall flavor of the Qingyang douchi. There were four main flavor compounds (ROAV ≥ 1) in Qingyang douchi, namely, 1-octen-3-ol, 2,5-dimethylpyrazine, 2-pentylfuran, and 2,4,5-trimethylthiazole. There were nine substances (0.1 ≤ ROAV < 1) that contributed to the modification of the overall flavor of Qingyang douchi, namely, 1-penten-3-ol, 2-butanone, 2-pentanone, 2-methylpyrazine, 2,6-dimethylpyrazine, 2,3-dimethylpyrazine, 2-ethylpyrazine, 2-ethyl-5-methylpyrazine, and 2,3,5-trimethylpyrazine.

The compounds playing a flavor role in Longnan douchi were detected less in Qingyang douchi, and compounds playing a flavor role in Qingyang douchi played a smaller role in Longnan douchi. According to the ROAVs of volatile flavor compounds, the key flavor compounds in Longnan douchi mainly consisted of esters, whereas the key flavor compounds in Qingyang douchi mainly consisted of pyrazines.

Among them, 1-octen-3-ol was the main flavor compound shared by the two types of douchi; it is a typical bean flavor compound with a mushroom flavor [33]. Ethyl isobutyrate has a strawberry-like odor [28], ethyl butyrate has a pineapple flavor and a floral aroma [29], isoamyl acetate has a banana flavor [30], ethyl hexanoate has an apple-like flavor [31], 2,5-dimethylpyrazine has a baking flavor [34], 2-pentylfuran has a legume and grassy flavor [35], and 2,4,5-trimethylthiazole has an earthy flavor [36]. These flavor compounds are usually produced by the metabolism of microorganisms in douchi during the fermentation process or produced by the degradation of starch substances in the raw materials by microorganisms during the glycolysis process. Together, these flavor compounds constitute the unique flavor of douchi.

### 3.3. Major Microorganismsin the Two Types of Douchi

To investigate the variations in the microbial communities in the two types of douchi, we assessed α-diversity using Chao1, Shannon, and Simpson indices. The bacterial diversity was significantly higher in QY than in LN douchi (*p* < 0.05) (Figure 1A–C). In contrast, the diversity of fungal communities was significantly higher in LN than in QY douchi (*p* < 0.05) (Figure 2A–C). The total number of bacterial ASVs in the LN and QY samples was 97 and 131, respectively. Among them, 10 ASVs (4.58%) were shared between them, whereas LN and QY had 87 and 121 unique ASVs, respectively (Figure 1D). As for fungi, the LN and QY samples had 471 and 203 ASVs, respectively. Of these, 22 ASVs (3.37%) were shared between them, whereas 449 were unique to LN, and 181 to QY (Figure 2D).

In terms of the bacterial community at the phylum level, Firmicutes was the most abundant taxon, accounting for >69% (Figure 1E). In samples of the LN douchi, four genera (*Bacillus*, *Ignatzschineria*, *Proteus*, and *Providencia*) were dominant, with relative abundances of 62.76, 8.65, 8.21, and 6.43%, respectively. Three dominant genera (*Carnobacterium*, *Ignatzschineria*, and *Bacillus*) were found in the QY douchi, with relative abundances of 51.14, 27.79, and 10.17%, respectively (Figure 1F). Ascomycota and Basidiomycota were the predominant phyla in the fungal community, accounting for more than 90% of the total abundance (Figure 2E). In samples of the LN douchi, three fungal genera (*Pichia*, *Candida*, and *Rhodosporidium*) were dominant, with relative abundances of 64.53, 16.93, and 9.31%, respectively. *Pichia*, the only dominant fungal genus, was observed in the QY douchi, with a relative abundance of 93.83% (Figure 2F).

### 3.4. Correlation Between Major Microorganisms and Flavor Compounds in the Two Types of Douchi

The intergroup correlation of dominant genera and the relative contents of main flavor compounds were determined for the two types of douchi. The interaction between microorganisms and flavor compounds in Longnan douchi is shown in Figure 3A. There was a significant negative correlation between *Bacillus* and ethyl hexanoate (*p* < 0.05), *Pichia* showed a significant negative correlation with ethyl isobutyrate (*p* < 0.05), *Candida* showed a highly significant negative correlation with ethyl butyrate (*p* < 0.01), and *Rhodosporidium* was significantly positively correlated with ethyl butyrate (*p* < 0.05). The interaction between microorganisms and flavor compounds in Qingyang douchi is shown in Figure 3B. *Carnobacterium* showed a significant positive correlation with 2,4,5-trimethylthiazole (*p* < 0.05).

In Longnan douchi, *Rhodosporidium* was the main contributor to the production of flavor compounds, whereas *Carnobacterium* was the main contributor to the flavor in Qingyang douchi. It has been reported that *Rhodosporidium toruloides* can produce lipids and sugar alcohols commonly used as natural sweeteners from galactose. This may be closely related to the production of sweetness in douchi [37]. *Carnobacterium* is able to convert glucose into lactic acid, and this group of microorganisms play an important role in the improvement of douchi flavor and nutritional value [38]. In Longnan douchi, *Bacillus*, *Pichia*, and *Candida* play an inhibitory role in the formation of flavor compounds, which is different from the results of the previous study, in which *Bacillus*, *Pichia*, and *Candida* were found in 10 commercial douchi to produce enzymes or flavor substances that play an important role in the flavor composition of douchi [19]. This may be due to the different fermentation processes and conditions of different douchi.

Therefore, the main reason for the great difference in flavor between the two types of douchi was that the microbial communities were different, leading to the formation of the final flavor through the growth and metabolism of microorganisms, which is consistent with the research conclusion of He et al. [1].

## 4. Conclusions

In this study, we systematically compared and analyzed the nutrient composition, volatile flavor compounds, and correlation between the flavor and microorganisms of Longnan and Qingyang douchi. We found a significant difference in the quality of the two types of douchi. Overall, the basic nutritional indices of Longnan douchi are higher than those of Qingyang douchi, and the flavor components are richer. The correlation analysis showed that microorganisms correlated with flavor compounds are mostly found in Longnan douchi, and some microorganisms exist in Longnan douchi that have an inhibitory effect on flavor production. Therefore, it can be inferred that the differences in nutrition and flavor between Longnan and Qingyang douchi may be due to the differences in microbial communities caused by the different production processes and environmental conditions of the two types of douchi. This study systematically elucidated the flavor and microbial correlation of Longnan and Qingyang douchi. Our results can serve as a reference for the establishment of a douchi fermentation system and improvement in douchi quality.

## Figures and Tables

**Figure 1 foods-13-03519-f001:**
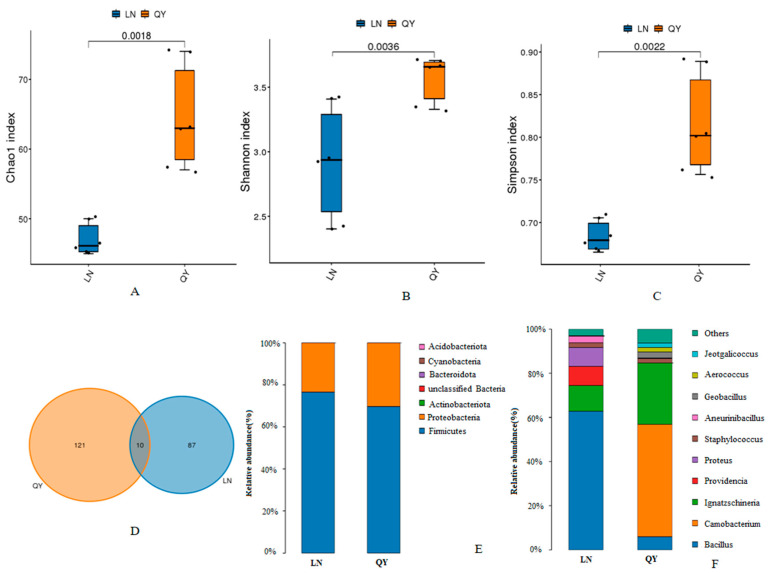
Alpha diversity assessed with Chao1 index (**A**), Shannon index (**B**), and Simpson index (**C**) of bacterial taxon in different groups. Venn plot of bacterial ASV number from different groups (**D**). Relative abundance of bacterial taxon at phylum (**E**) and genus (**F**) levels in different groups.

**Figure 2 foods-13-03519-f002:**
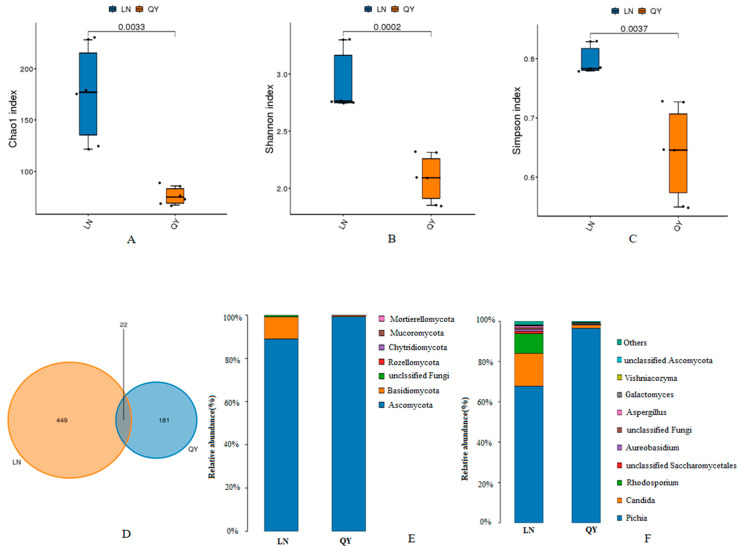
Alpha diversity assessed with Chao1 index (**A**), Shannon index (**B**), and Simpson index (**C**) of fungal taxon in two types of douchi. Venn plot of fungal ASV number from two types of douchi (**D**). Relative abundance of fungal taxon at phylum (**E**) and genus (**F**) levels in two types of douchi.

**Figure 3 foods-13-03519-f003:**
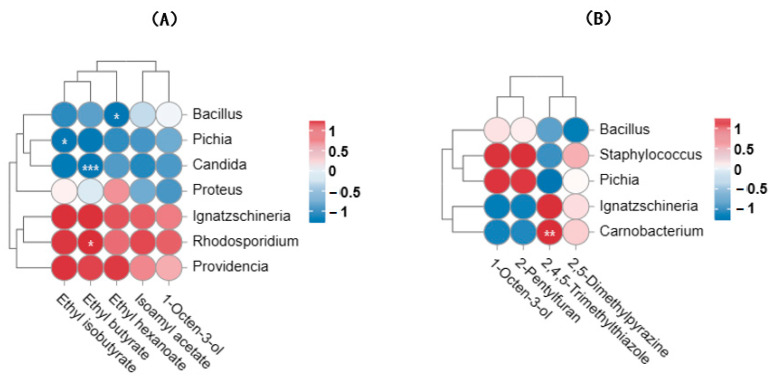
Correlation analysis of major microorganisms and main flavor compounds in (**A**) Longnan and (**B**) Qingyang douchi. Red indicates positive correlation, while blue indicates negative correlation (the deeper the color, the stronger the correlation). *** *p* < 0.001; ** *p* < 0.01; * *p* < 0.05.

**Table 1 foods-13-03519-t001:** Test results of basic nutritional components in Longnan and Qingyang douchi.

Physicochemical Indicators	LN Douchi	QY Douchi
Moisture (%)	33.96 ± 0.44 ^b^	37.24 ± 0.47 ^a^
Fat (%)	2.09 ± 0.01 ^b^	9.19 ± 0.05 ^a^
Amino acid nitrogen (g/100 g)	1.16 ± 0.01 ^a^	0.78 ± 0.00 ^b^
Salt (g/100 g)	5.92 ± 0.21 ^a^	4.19 ± 0.13 ^b^
Total acid (g/100 g)	1.98 ± 0.02 ^a^	2.04 ± 0.02 ^a^
Protein (g/100 g)	34.14 ± 0.26 ^a^	28.49 ± 0.16 ^b^

Note: Letters indicate Duncan’s pairwise differences among different samples (*p* < 0.05).

**Table 2 foods-13-03519-t002:** Composition and content of FFAs in Longnan and Qingyang douchi.

FFAs	Content (%)	
	LN Douchi	QY Douchi
Saturated Fatty Acids (SFAs)		
Myristic acid (C14:0)	0.04 ± 0.00 ^b^	0.07 ± 0.01 ^a^
Palmitic acid (C16:0)	10.65 ± 0.10 ^a^	9.63 ± 0.14 ^b^
Heptadecanoic acid (C17:0)	0.09 ± 0.00 ^a^	0.08 ± 0.00 ^b^
Stearic acid (C18:0)	3.41 ± 0.02 ^a^	3.52 ± 0.03 ^a^
Arachidic acid (C20:0)	0.21 ± 0.02 ^b^	0.29 ± 0.01 ^a^
Behenic acid (C22:0)	0.28 ± 0.02 ^a^	0.30 ± 0.01 ^a^
Lignoceric acid (C24:0)	0.06 ± 0.01 ^b^	0.11 ± 0.01 ^a^
Total	14.74 ± 0.14	14.00 ± 0.21
Monounsaturated Fatty Acids (MUFAs)		
Heptacenoic acid (C17:1)	0.04 ± 0.00 ^a^	0.03 ± 0.00 ^b^
Oleic acid (C18:1n-9)	18.80 ± 0.04 ^b^	20.01 ± 0.13 ^a^
*cis*-11-Eicosenoic acid (C20:1)	0.03 ± 0.00 ^a^	0.03 ± 0.01 ^a^
Erucic acid (C20:1)	0.07 ± 0.01 ^a^	0.02 ± 0.00 ^b^
Tetradecanoic acid (C24:1)	0.26 ± 0.06	-
Total	19.20 ± 0.11	20.09 ± 0.11
Polyunsaturated Fatty Acids (PUFAs)		
Linoleic acid (C18:2)	56.79 ± 0.08 ^a^	55.54 ± 0.07 ^b^
Linolenic acid (C18:3)	9.25 ± 0.03 ^b^	10.32 ± 0.06 ^a^
Docosadienoic acid (C22:2)	0.08 ± 0.01	-
*cis*-11,14-Eicosadienoic acid (C20:2)	0.02 ± 0.00 ^a^	0.02 ± 0.00 ^a^
Total	66.14 ± 0.12	65.88 ± 0.13
PUFA/SFA	4.49	4.71

Note: Letters indicate Duncan’s pairwise differences among different samples (*p* < 0.05).

**Table 3 foods-13-03519-t003:** Composition and content of FAAs in Longnan and Qingyang douchi (g/100 g).

FAAs		Content (g/100 g)	
		LN Douchi	QY Douchi
Essential amino acids	Threonine (Thr)	0.58 ± 0.02 ^a^	0.24 ± 0.02 ^b^
	Lysine (Lys)	0.82 ± 0.00 ^a^	0.60 ± 0.02 ^b^
	Isoleucine (Ile)	0.80 ± 0.01 ^a^	0.43 ± 0.00 ^b^
	Leucine (Leu)	1.72 ± 0.02 ^a^	0.91 ± 0.02 ^b^
	Phenylalanine (Phe)	1.19 ± 0.01 ^a^	0.66 ± 0.00 ^b^
	Valine (Val)	0.74 ± 0.01 ^a^	0.45 ± 0.01 ^b^
Non-essential amino acids	Alanine (Ala)	1.15 ± 0.08 ^a^	0.65 ± 0.01 ^b^
	Histidine (His)	0.56 ± 0.02 ^a^	0.41 ± 0.00 ^b^
	Proline (Pro)	2.11 ± 0.08 ^a^	1.68 ± 0.02 ^b^
	Glutamate (Glu)	6.13 ± 0.11 ^a^	4.81 ± 0.02 ^b^
	Tyrosine (Tyr)	0.22 ± 0.02 ^b^	0.35 ± 0.01 ^a^
	Aspartate (Asp)	0.84 ± 0.02 ^a^	0.64 ± 0.01 ^b^
	Cystine (Cys)	0.32 ± 0.02 ^a^	0.11 ± 0.01 ^b^
	Serine (Ser)	0.62 ± 0.02 ^a^	0.24 ± 0.01 ^b^
	Glycine (Gly)	0.63 ± 0.02 ^a^	0.21 ± 0.01 ^b^
	Arginine (Arg)	0.86 ± 0.01 ^a^	0.47 ± 0.01 ^b^
Total		19.31 ± 0.86	12.58 ± 0.18

Note: Letters indicate Duncan’s pairwise differences among different samples (*p* < 0.05).

**Table 4 foods-13-03519-t004:** Results of volatile flavor compound analysis in Longnan and Qingyang douchi.

No.	Category	Compound Name	Relative Content (%)	
			LN Douchi	QY Douchi
	**Esters**			
1		Ethyl acetate	3.93 ± 0.29	-
2		Ethyl isobutyrate	4.36 ± 0.31	-
3		Ethyl Butyrate	4.90 ± 0.30	-
4		Ethyl 2-methylbutyrate	2.59 ± 0.14	-
5		Ethyl isovalerate	8.37 ± 0.47	-
6		Isoamyl acetate	1.35 ± 0.07	-
7		Ethyl hexanoate	2.57 ± 0.13	-
8		Isoamyl butyrate	0.27 ± 0.04	-
9		2-Methylbutyrate-3-methylbutyl ester	0.32 ± 0.03	-
10		3-Methyl-2-methylbutyl butyrate	0.47 ± 0.07	-
11		Ethyl heptanoate	0.36 ± 0.04	-
12		Ethyl caprylate	0.29 ± 0.02 ^a^	0.05 ± 0.01 ^b^
13		Linalyl acetate	1.89 ± 0.10	-
14		Methyl benzoate	0.55 ± 0.02 ^a^	0.12 ± 0.01 ^b^
15		γ-butyrolactone	0.10 ± 0.06	-
	**Alcohols**			
16		Ethanol	5.75 ± 0.52	-
17		2-Methyl-1-propanol	0.19 ± 0.04	-
18		1-Penten-3-ol	-	0.13 ± 0.01
19		Isoamyl alcohol	3.17 ± 0.08 ^b^	7.87 ± 0.03 ^a^
20		2-Heptanol	0.09 ± 0.01	-
21		3-Nonyl alcohol	-	0.30 ± 0.02
22		Hexanol	0.11 ± 0.01	-
23		3-Octanol	0.37 ± 0.02	-
24		2-Octanol	0.02 ± 0.01	-
25		Anti- α,α- 5-Trimethyl-5-vinyltetrahydro-2-furan methanol	0.11 ± 0.02	-
26		1-octen-3-ol	1.38 ± 0.02 ^a^	0.18 ± 0.01 ^b^
27		3-Methylcyclohexanol	-	0.06 ± 0.00
28		1-Octanol	0.09 ± 0.01	-
29		2,3-butanediol	1.22 ± 0.07	-
30		4-Terpeneol	1.31 ± 0.04	-
31		2-ethylhexanol	-	0.18 ± 0.01
32		furfuryl alcohol	0.48 ± 0.01	-
	**Ketones**			
33		Acetone	-	1.99 ± 0.53
34		2-Butanone	-	9.27 ± 1.14
35		2-Pentanone	-	8.54 ± 0.41
36		4-Methyl-2-pentanone	-	0.29 ± 0.15
37		3-Methyl-2-pentanone	-	3.01 ± 0.35
38		2,3-Pentanedione	-	0.05 ± 0.00
39		2-Heptanone	-	0.79 ± 0.07
40		Cyclopentanone	-	1.41 ± 0.29
41		2-Octanone	0.07 ± 0.01	-
42		5-Methyl-3-heptanone	-	1.15 ± 0.07
43		3-Hydroxy-2-butanone	0.29 ± 0.01 ^b^	0.91 ± 0.01 ^a^
44		3-Octanone	1.11 ± 0.07	-
45		2,2,5-Trimethylcyclopentanone	-	0.13 ± 0.01
46		2-Hydroxy-3-pentanone	-	0.15 ± 0.03
47		2-Nonanone	-	0.43 ± 0.02
48		4-Cyclohepten-1-one	-	2.76 ± 0.13
49		3,4,4-Trimethylcyclopent-2-enone	-	0.27 ± 0.02
50		Acetophenone	0.18 ± 0.01 ^a^	0.25 ± 0.01 ^a^
	**Aldehydes**			
51		Furfural	0.31 ± 0.05	-
52		Benzaldehyde	0.70 ± 0.09	-
53		5-Methylfurfural	0.19 ± 0.01	-
	**Terpenoids**			
54		Laurylene	0.83 ± 0.09	-
55		Limonene	1.51 ± 0.41	-
56		*γ*- Terpinene	0.07 ± 0.03	-
57		*α*-Pinene	0.05 ± 0.01	-
	**Pyrazines**			
58		2-Methylpyrazine	-	3.42 ± 0.20
59		2,5-Dimethylpyrazine	2.33 ± 0.03 ^b^	26.77 ± 1.26 ^a^
60		2,6-Dimethylpyrazine	1.05 ± 0.03 ^b^	2.32 ± 0.01 ^a^
61		2,3-Dimethylpyrazine	1.70 ± 0.03 ^a^	0.49 ± 0.03 ^b^
62		2-Ethylpyrazine	-	0.17 ± 0.01
63		2-ethyl-5-methylpyrazine	-	0.15 ± 0.01
64		2,3,5-trimethylpyrazine	6.80 ± 0.19 ^a^	4.71 ± 0.25 ^b^
65		3-ethyl-2,5-methylpyrazine	-	0.17 ± 0.02
66		2,3,5-trimethyl-6-ethylpyrazine	0.27 ± 0.02	-
	**Furans**			
67		2-Pentylfuran	-	0.61 ± 0.09
68		2-Acetylfuran	-	0.11 ± 0.01
	**Thiazoles**			
69		2,4,5-Trimethylthiazole	0.96 ± 0.01 ^a^	0.11 ± 0.02 ^b^
	**Pyridines**			
70		2,3,4,5-Tetrahydropyridine	-	0.33 ± 0.02
	**Pyrroles**			
71		N-methylpyrrole	-	1.23 ± 0.06
	**Aromatics**			
72		Toluene	-	0.59 ± 0.03
	**Others**			
73		Acetonitrile	-	0.16 ± 0.09
74		Dimethyl disulfide	-	1.19 ± 0.15
75		(2E,6E)-4,5-Dimethyl-2,6-octadiene	0.11 ± 0.02	-
76		Dimethyl trisulfide	0.14 ± 0.01	-
77		Ligustrazine	13.40 ± 0.35 ^a^	2.44 ± 0.36 ^b^

Note: “-” not perceived; letters indicate Duncan’s pairwise differences among different samples (*p* < 0.05).

**Table 5 foods-13-03519-t005:** ROAVs of volatile flavor compounds in Longnan and Qingyang douchi.

Category	Compound Name	Aroma Descriptors	Aroma Threshold (μg/kg) [27]	ROAVs	
				LN Douchi	QY Douchi
**Esters**					
	Ethyl isobutyrate	strawberry-like odor [28]	0.1	100.00	-
	Ethyl butyrate	pineapple flavor and a floral aroma [29]	1	11.24	-
	Isoamyl acetate	banana [30]	2	1.55	-
	Ethyl hexanoate	apple [31]	1	5.89	-
	Ethyl heptanoate	fruity, wine-like [32]	2.2	0.38	-
**Alcohols**					
	1-Octen-3-ol	mushroom [33]	1	3.17	100.00
	1-Penten-3-ol	vegetable, fruity [32]	400	-	0.18
	Hexanol	sweet, fruity, green, apple [32]	250	0.00	-
**Ketones**					
	Acetone	slightly sweet, pungent [32]	500,000	-	0.00
	2-Butanone	coffee, banana [32]	50,000	-	0.10
	2-Pentanone	sweet, fruity [32]	70,000	-	0.07
	2-Octanone	sweet floral, green, apple [32]	50	0.00	-
	3-Octanone	mushroom, green, vegetable, fruity [32]	28	0.09	-
**Aldehydes**					
	Furfural	bread, caramel, nuts [32]	3000	0.00	-
**Pyrazines**					
	2-Methylpyrazine	nuts, baking flavor [32]	6000	-	0.32
	2,5-Dimethylpyrazine	a baking flavor [34]	1800	0.00	8.26
	2,6-Dimethylpyrazine	fried and roasted flavor, nuts, cocoa [32]	1500	0.00	0.86
	2,3-Dimethylpyrazine	roast, cocoa, nuts [32]	2500	0.00	0.11
	2-Ethylpyrazine	roast, cocoa, nuts [32]	6000	-	0.02
	2-ethyl-5-methylpyrazine	roast, cocoa, nuts [32]	100	-	0.83
	2,3,5-Trimethylpyrazine	roast, cocoa, nuts [32]	9000	0.00	0.29
**Furans**					
	2-Pentylfuran	has a legume and grassy flavor [35]	6	-	56.48
**Thiazoles**					
	2,4,5-Trimethylthiazole	earthy [36]	50	0.04	1.22

Note: “-” not perceived.

## Data Availability

The original contributions presented in the study are included in the article/Appendix A, further inquiries can be directed to the corresponding authors.

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
