# Peer review of "Nutrition, Flavor, and Microbial Communities of Two Traditional Bacterial Douchi from Gansu, China"

_foods, 2024, doi:10.3390/foods13213519_

Round 1
Reviewer 1 Report
Comments and Suggestions for Authors
The authors describe their investigation of two different types of traditional 2 bacterial douchi. The authors compared four different aspects of these douchi, the basic nutrient composition, the free amino acids, the aroma compounds, and the microbiology. Overall the manuscript is clearly written and results are primarily supported by data and literature. While the experiments are clearly communicated and connected to highlight the differences of the two different douchi. I have some specific comments and questions.
What is the difference in the information in section 2.2.2 and 2.2.3. The two sections are labeled exactly the same but reference different method numbers. Please distinguish what is actually being measured in each.
line 105: is the (3.0) shown here a mass?
line 167-171: I am not sure that these two statements are completely accurate. What is this based on? Could some of these basic nutrient composition be a function of differences in the original bean products?
Author Response
Comments 1: What is the difference in the information in section 2.2.2 and 2.2.3. The two sections are labeled exactly the same but reference different method numbers. Please distinguish what is actually being measured in each.
Response: Thank you for pointing this out. The subsections are not repeated and we have added the full names in the revised manuscript.
Comments 2: line 105: is the (3.0) shown here a mass?
Response: Thank you for pointing this out. We agree with this comment. Therefore, we have added the unit in the revised manuscript.
Comments 3: line 167-171: I am not sure that these two statements are completely accurate. What is this based on? Could some of these basic nutrient composition be a function of differences in the original bean products?
Response: Thank you for pointing this out. We agree with this comment. Therefore, we have deleted these sentences.
Reviewer 2 Report
Comments and Suggestions for Authors
The manuscript explores the nutrition, flavor, and the correlation between flavor and microorganisms in two traditional bacterial douchi from Gansu Province in northwest China. However, the main contributions of the article are not clearly stated or discussed. Additionally, the statistical methods used to compare the compositions of the two samples are inadequately described, some statements are speculative, and the quality of several figures is very poor.
While the paper may hold potential interest, it has several shortcomings, as outlined below.
1. Introduction section
Line 65: What aspects of the bacterial bacterial-type douchi were studied in the few reports on this type of products? What are these few reports? Please cite these reports.
What new contributions does this work with respect to these previous reports?
2. Materials and methods section
2.1. Lines 78-82: Please provide a more detailed description of these methods to facilitate replication by other researchers.
2.2. Lines 83 and 85: These subsections are repeated.
2.3. Lines 84 and 86: While these subsections are repeated, the referenced analytical methods are different. Please, correct this.
2.4. Lines 146-150: How many replicates were used in the different analysis? How were the mean concentrations of nutrients in the two types of douchi compared? Please explain.
3. Results and analysis section
3.1. Lines 155-171: Why did the authors not include the composition of the substrates used for douchi production? It is well known that substrate composition is a key variable that significantly influences the composition of the fermented products.
3.2. Table 1. This Table should be placed after its first mention in the text.
3.3. Line 164: Which other transformation actions are the authors referring to?
3.4. Lines 167-168: Which microorganisms are commonly used to produce the two douchi products? What types of enzymes do they produce?
3.5. Lines 169-171: This statement is speculative because the authors did not provide references or evidences to support it.
3.6. Table 2: What statistical method was used to compare the compositions of Longnan and Qingyang douchi? This Table should be placed after its first mention in the text.
3.7. Table 3: Table 2: What statistical method was used to compare the FAAs content in Longnan and Qingyang douchi? This Table should be placed after its first mention in the text.
3.8. Table 5: Please provide the references from which the aroma threshold of the different volatile compounds detected in the two samples were obtained.
3.9. Table 5: Please include the aroma descriptors and the corresponding references for the different volatile compounds detected in the two samples.
3.10. The quality of Figures 1 and 2 should be significantly improved.
4. References section
Some references are cited with their full names while others are cited with abbreviated names. Please correct this inconsistency.
Author Response
Comments 1: Line 65: What aspects of the bacterial bacterial-type douchi were studied in the few reports on this type of products? What are these few reports? Please cite these reports.
What new contributions does this work with respect to these previous reports?
Response: Thank you for pointing this out. We agree with this comment. Therefore, we have added the references about bacterial-type douchi in the revised manuscript.
- FanJ.F.; ZhangY.Y.; ChangX.J.; SaitoM.; LiZ.G.Changes in the radical scavenging activity of bacterial-type Douchi, a traditional fermented soybean product, during the primary fermentation process[J]. Biosci Biotech Bioch.2009,73, 2749-2753.
10.Wang, N.;Chen, W.X.;Zhang, W.P.;Zhang, J.S.Community diversity of different bacterial-type of Guizhou Douchi using high throughput sequencing[J].Food and Fermentation Industries.2022,48, 85-90,104. (in chinese)
Comments 2: 2.1. Lines 78-82: Please provide a more detailed description of these methods to facilitate replication by other researchers.
Response: Thank you for pointing this out. We agree with this comment. Due to concerns about higher duplication rate, we provide website for downloading these standards. These Chinese Standards can be found online at foodmate(http://down.foodmate.net/standard/).
Comments 3: 2.2. Lines 83 and 85: These subsections are repeated.2.3. Lines 84 and 86: While these subsections are repeated, the referenced analytical methods are different. Please, correct this.
Response: Thank you for pointing this out. The subsections are not repeated and we have added the full names of FFAs and FAAs in the revised manuscript.
Comments 4: 2.4. Lines 146-150: How many replicates were used in the different analysis? How were the mean concentrations of nutrients in the two types of douchi compared? Please explain.
Response: Thank you for pointing this out. There were three replicates in each sample. We used the t-test to compare the mean values.
Comments 5: 3.1. Lines 155-171: Why did the authors not include the composition of the substrates used for douchi production? It is well known that substrate composition is a key variable that significantly influences the composition of the fermented products.
Response: Thank you for pointing this out. We agree that substrate composition is a key variable that significantly influences the composition of the fermented products. They all use soybean as raw material to produce douchi, but we have not collected relevant samples in this experiment. We will conduct related experiments in follow-up studies.
Comments 6: 3.2. Table 1. This Table should be placed after its first mention in the text.
Response: Thank you for pointing this out. We agree with this comment. Therefore, we have made changes for Table 1.
Comments 7: 3.3. Line 164: Which other transformation actions are the authors referring to?
Response: Thank you for pointing this out. Mainly refers to some chemical reactions such as Maillard reaction.
Comments 8: 3.4. Lines 167-168: Which microorganisms are commonly used to produce the two douchi products? What types of enzymes do they produce?
Response: Thank you for pointing this out. These statements are inaccurate and have been removed from the revised version.
Comments 9: 3.5. Lines 169-171: This statement is speculative because the authors did not provide references or evidences to support it.
Response: Thank you for pointing this out. We agree with this comment. Therefore, we have deleted these sentences.
Comments 10: 3.6. Table 2: What statistical method was used to compare the compositions of Longnan and Qingyang douchi? This Table should be placed after its first mention in the text.
Response: Thank you for pointing this out. We used the t-test to compare the mean values. Therefore, we have made changes for Table 2 and have added note under the table.
Comments 11: 3.7. Table 3: Table 2: What statistical method was used to compare the FAAs content in Longnan and Qingyang douchi? This Table should be placed after its first mention in the text.
Response: Thank you for pointing this out. We used the t-test to compare the mean values. Therefore, we have made changes for Table 3 and have added note under the table.
Comments 12: 3.8. Table 5: Please provide the references from which the aroma threshold of the different volatile compounds detected in the two samples were obtained.
Response: Thank you for pointing this out. We agree with this comment. Therefore, we have added the references in the revised manuscript.
27.Van Gemert, L. J. Odour thresholds: Compilations of odour threshold values in air, water and other media (Edition 2011).; Publisher: Oliemans Punter & Partners BV, Zeist, The Netherlands, 2011.
Comments 13: 3.9. Table 5: Please include the aroma descriptors and the corresponding references for the different volatile compounds detected in the two samples.
Response: Thank you for pointing this out. We agree with this comment. Therefore, we have added the references in the revised manuscript.
Comments 14: 3.10. The quality of Figures 1 and 2 should be significantly improved.
Response: Thank you for pointing this out. We agree with this comment. Therefore, we have replaced the two figures in the revised manuscript.
Comments 15: Some references are cited with their full names while others are cited with abbreviated names. Please correct this inconsistency.
Response: Thank you for pointing this out. We agree with this comment. Therefore, we have revised the references in the manuscript.